# Identification and characterization of novel alphacoronaviruses in *Tadarida brasiliensis* (Chiroptera, Molossidae) from Argentina: insights into recombination as a mechanism favoring bat coronavirus cross-species transmission

Agustina Cerri,[1] Elisa M. Bolatti,[1,2,3] Tomaz M. Zorec,[4] Maria E. Montani,[3,5,6] Agustina Rimondi,[7,8] Lea Hosnjak,[4] Pablo E. Casal,[9] Violeta Di Domenica,[1,3] Ruben M. Barquez,[3,6] Mario Poljak,[4] Adriana A. Giri[1,2]

**ABSTRACT** Bats are reservoirs of various coronaviruses that can jump between bat species or other mammalian hosts, including humans. This article explores coronavirus infection in three bat species (*Tadarida brasiliensis, Eumops bonariensis*, and *Molossus molossus*) of the family Molossidae from Argentina using whole viral metagenome analysis. Fecal samples of 47 bats from three semiurban or highly urbanized areas of the province of Santa Fe were investigated. After viral particle enrichment, total RNA was sequenced using the Illumina NextSeq 550 instrument; the reads were assembled into contigs and taxonomically and phylogenetically analyzed. Three novel complete Alphacoronavirus (AlphaCoV) genomes (Tb1–3) and two partial sequences were identified in *T. brasiliensis* (Tb4–5), and an additional four partial sequences were identified in *M. molossus* (Mm1–4). Phylogenomic analysis showed that the novel AlphaCoV clustered in two different lineages distinct from the 15 officially recognized AlphaCoV subgenera. Tb2 and Tb3 isolates appeared to be variants of the same virus, probably involved in a persistent infectious cycle within the *T. brasiliensis* colony. Using recombination analysis, we detected a statistically significant event in Spike gene, which was reinforced by phylogenetic tree incongruence analysis, involving novel Tb1 and AlphaCoVs identified in *Eptesicus fuscus* (family Vespertilionidae) from the U.S. The putative recombinant region is in the S1 subdomain of the Spike gene, encompassing the potential receptor-binding domain of AlphaCoVs. This study reports the first AlphaCoV genomes in molossids from the Americas and provides new insights into recombination as an important mode of evolution of coronaviruses involved in cross-species transmission.

**IMPORTANCE** This study generated three novel complete AlphaCoV genomes (Tb1, Tb2, and Tb3 isolates) identified in individuals of *Tadarida brasiliensis* from Argentina, which showed two different evolutionary patterns and are the first to be reported in the family Molossidae in the Americas. The novel Tb1 isolate was found to be involved in a putative recombination event with alphacoronaviruses identified in bats of the genus *Eptesicus* from the U.S., whereas isolates Tb2 and Tb3 were found in different collection seasons and might be involved in persistent viral infections in the bat colony. These findings contribute to our knowledge of the global diversity of bat coronaviruses in poorly studied species and highlight the different evolutionary aspects of AlphaCoVs circulating in bat populations in Argentina.

**KEYWORDS** bats, genus alphacoronavirus, novel genomes, recombination, Molossidae, cross-species transmission, Americas

Address correspondence to Elisa M. Bolatti, bolatti@ibr-conicet.gov.ar, Mario Poljak, mario.poljak@mf.uni-lj.si, or Adriana A. Giri, giri@ibr-conicet.gov.ar.

The authors declare no conflict of interest.

See the funding table on p. 16.

Coronaviruses (order: *Nidovirales*; family: *Coronaviridae*) are single-stranded positive-sense RNA viruses with the largest non-segmented RNA viral genomes among human viruses, ranging between 16 and 31 kb. Due to their large genome size, high recombination rates, and genomic plasticity, coronaviruses are able to jump cross-species barriers and rapidly adapt to new hosts (1). Based on their phylogenetic relationships and genomic structures, members of the subfamily *Orthocoronavirinae* have been divided into four genera: *Alphacoronavirus* (AlphaCoV) and *Betacoronavirus* (BetaCoV), which have been associated with infections in mammals, and *Gammacoronavirus* (GammaCoV) and *Deltacoronavirus* (DeltaCoV), which appear to mainly infect birds (2, 3).

Some AlphaCoVs have been recognized as causative agents of mild respiratory syndromes in immunocompetent humans (HCoV-NL63 and HCoV-229E) and of serious respiratory diseases and bowel disorders in livestock, such as transmissible gastroenteritis coronavirus, porcine epidemic diarrhea virus, and swine acute diarrhea syndrome, which are responsible for pandemics in pigs and cause significant economic losses (2, 4). On the other hand, BetaCoVs cause serious diseases in humans, such as severe acute respiratory syndrome (caused by SARS-CoV), Middle East respiratory syndrome (caused by MERS-CoV), and coronavirus disease 2019 (COVID-19) (caused by SARS-CoV-2) (4). A cumulative body of research on coronaviruses has shown that most AlphaCoVs and BetaCoVs infecting humans jumped and naturalized from animal hosts (2, 5). Bats (order Chiroptera) have often been reported as the source of zoonotic spillovers due to their characteristics, such as a relatively long lifespan, capacity of flight, high metabolic rates, and gregarious social behavior, which make them suitable for hosting and spreading a wide variety of viruses (6).

According to the database of zoonotic and vector-borne viruses, coronaviruses represent about 41% ($n = 7,446$) of all viral sequences globally reported in bats ($n = 18,152$), with AlphaCoVs ($n = 3,104$) appearing to be the most widespread viruses in these mammals (ZOVER, http://www.mgc.ac.cn/cgi-bin/ZOVER/mainTable.cgi?db=bat, accessed on 27 March 2023) (7, 8). Due to the geographic origin of the most prominent zoonotic coronaviruses, the main information comes from studies performed on Old World bat species. In contrast, so far, less research has been conducted in the Americas because only a total of 340 coronaviral sequences, mostly of the genus AlphaCoV, from 39 species of bats from this geographical region belonging to four families (Molossidae, Mormoopidae, Phyllostomidae, and Vespertilionidae), have been identified (ZOVER, http://www.mgc.ac.cn/cgi-bin/ZOVER/mainTable.cgi?db=bat, accessed on 27 March 2023). Among these, only seven AlphaCoV complete genomes have been characterized in species of bats of the family Vespertilionidae in North America (the U.S.) (9, 10) and in vampire bats (family Phyllostomidae) in South America (Peru) (11). On the other hand, Molossidae (P. Gervais, 1856) is a cosmopolitan family of bats, widely distributed throughout the world, and often found in tropical and subtropical regions. Currently, it includes two subfamilies: Tomopeatinae, monotypic and endemic to Peru, and Molossinae, which is cosmopolitan (12, 13), with seven genera and 20 species recorded in Argentina (12, 14). Therefore, to understand virus ecology within bat reservoirs, anticipate zoonotic risk, and accelerate the identification of reservoir hosts following the emergence of the disease, greater efforts are needed for intensive surveillance of coronaviruses in geographical areas that are understudied.

In line with this, in a previous study, we identified traces of coronavirus RNA in oral/anal samples of *Tadarida brasiliensis* (I. Geoffroy Saint-Hilaire, 1824), an arthropodophagous species of bats of the family Molossidae (15). To fully characterize the coronaviruses identified and to explore other potential transmission routes, here we analyze a total of 47 fecal samples from three species of bats of the family Molossidae that inhabit three semiurban or highly urbanized areas of the province of Santa Fe (Argentina). Using whole viral metagenome analysis, we characterized three novel complete genomes and six partial sequences of AlphaCoVs in *T. brasiliensis* and *Molossus molossus* (Pallas, 1766), which, to our knowledge, represent their first description in Molossidae bats

from the Americas. In addition, we detected a recombination event in the Spike gene, which may be involved in past cross-species transmission infections between molossid and vespertilionid bats. Our findings increase the knowledge of coronavirus infection dynamics among bats living in close contact with humans in the Americas and highlight the need for systematic active surveillance of AlphaCoVs as potential human pathogens in Argentina.

## MATERIALS AND METHODS

### Study area, sample collection, and ethics statement

Fecal samples were collected from three bat species of the family Molossidae—*T. brasiliensis*, *Eumops bonariensis* (Peters, 1874), and *M. molossus*—at three different geographical sites in the province of Santa Fe (Argentina): downtown Rosario (*T. brasiliensis* maternal colony), Villarino Park in Zavalla [both sites described previously by Bolatti et al. (16)], and the Ecological Reserve of the National University of the Littoral in Santa Fe (31°38'10"S 60°40'31"W). Samples of *T. brasiliensis* were collected in 2016/2017 and 2017/2018, whereas samples of *E. bonariensis* and *M. molossus* were obtained in 2017 (Table 1).

Sample collection was performed as previously described (15, 16). Briefly, bats were manually captured from the walls or using mist nets and placed in individual cotton bags for the determination of their species based on external and cranial morphometric characteristics, reproductive condition, sex, and relative age. Fecal drops were collected from the individual bags using sterile cotton-tipped swabs, suspended in 1 mL of viral transport media, and stored at 4°C or on dry ice until further processing. Subsequently, the animals were rehydrated and released.

During this study, every effort was made to minimize animal disturbance and suffering; no breeding or pregnant female bats were captured, and no animals were harmed or required euthanasia. Sampling was carried out by trained professionals as approved by the Ministry of Environment of the Argentinian Province of Santa Fe (content numbers 519/17, 358, and 356) and the Animal Ethics Committee of the Faculty of Biochemical and Pharmaceutical Sciences of the National University of Rosario (consent number 6060/243).

### Sample processing and viral enrichment

Selected fecal samples from 47 adult individual bats were vortexed to completely resuspend the material into solution. Subsequently, Hank's balanced salt solution (HBSS) was added to each sample to reach 1 mL and further vortexed to create a less viscous solution. The suspensions were then centrifuged at 10,000 $\times$ $g$ for 2 min, and the supernatants were transferred to fresh tubes and pooled by bat species, collection date, and site (Table 1). Each pool was filtered through a 0.45-µm pore-size syringe filter (Fisher Scientific, Pittsburgh, PA) and then centrifuged at 50,000 $\times$ $g$ for 3 h at 10°C. Each pellet was then resuspended in 100 µL of HBSS and frozen at −80°C until further processing could be performed. Next, to reduce the amount of contaminating host RNA and DNA, each sample was treated with 14 U of DNase I [New England Biolabs (NEB), Ipswich, MA] and 20 U of RNase H (NEB), made up to a final volume of 140 µL in 10× DNase buffer (NEB) and nuclease-free water, and incubated at 37°C for 2 h. Later, total nucleic acids from viral particles were extracted using the Viral DNA/RNA Kit (Macherey-Nagel, Düren, Germany), and viral RNA and DNA were eluted to a final volume of 30 µL and stored at −80°C until further use.

### Library preparation and viral metagenome shotgun sequencing

First-strand cDNA synthesis was performed with SuperScript IV Reverse Transcriptase (Thermo Scientific, Waltham, MA) and random hexamers (NEB), following the manufacturer's recommendations. The synthesis of the second strand of cDNA was performed

**TABLE 1** Bat samples included in the present study by species, collection site, and date

| Source | Location | Collection date | Pool ID | Sample ID | Bat species | Gender |
|---|---|---|---|---|---|---|
| Bat colony | Rosario city | 11/28/17 | 1 | M119 | *Tadarida* | Female |
| | | 11/28/17 | | M120 | *brasiliensis* | Female |
| | | 11/28/17 | | M122 | | Female |
| | | 11/28/17 | | M123 | | Female |
| | | 11/28/17 | | M124 | | Female |
| | | 11/28/17 | | M125 | | Female |
| | | 11/28/17 | | M127 | | Female |
| | | 11/28/17 | | M128 | | Female |
| | | 11/28/17 | | M132 | | Female |
| | | 11/28/17 | | M133 | | Female |
| | | 12/13/16 | 2 | M02 | *Tadarida* | Female |
| | | 12/13/16 | | M04 | *brasiliensis* | Female |
| | | 12/13/16 | | M05 | | Female |
| | | 12/13/16 | | M07 | | Female |
| | | 12/13/16 | | M08 | | Female |
| | | 12/13/16 | | M09 | | Female |
| | | 12/13/16 | | M12 | | Female |
| | | 12/13/16 | | M13 | | Female |
| | | 12/13/16 | | M15 | | Female |
| | | 12/13/16 | | M19 | | Female |
| Individual bats | Villarino Park Zavalla city | 02/03/17 | 3 | M69 | *Eumops* | Male |
| | | 02/03/17 | | M70 | *bonariensis* | Female |
| | | 02/03/17 | | M71 | | Female |
| | | 02/03/17 | | M72 | | Male |
| | | 04/13/17 | | M102 | | Female |
| | | 04/13/17 | | M106 | | Female |
| | | 04/12/17 | | M92 | | Male |
| | | 04/12/17 | | M87 | | Female |
| | | 04/13/17 | | M97 | | Female |
| | | 04/13/17 | | M99 | | Male |
| Individual bats | Ecological Reserve, National University of the Littoral Santa Fe city | 03/17/17 | 4 | M80 | *Eumops* | Female |
| | | 03/17/17 | | M81 | *bonariensis* | Female |
| | | 03/17/17 | | M82 | | Female |
| | | 03/17/17 | | M83 | | Female |
| | | 03/17/17 | 5 | M76 | *Molossus* | Female |
| | | 03/17/17 | | M77 | *molossus* | Female |
| | | 03/18/17 | | M78 | | Female |
| | | 03/18/17 | | M79 | | Male |
| Individual bats | Villarino Park Zavalla city | 04/13/17 | 6 | M94 | *Molossus* | Female |
| | | 04/13/17 | | M108 | *molossus* | Female |
| | | 04/12/17 | | M85 | | Female |
| | | 04/12/17 | | M89 | | Male |
| | | 04/12/17 | | M90 | | Male |
| | | 04/13/17 | | M93 | | Female |
| | | 04/13/17 | | M95 | | Male |
| | | 04/13/17 | | M96 | | Female |
| | | 04/13/17 | | M103 | | Female |

using the Second Strand cDNA Synthesis Kit (Thermo Scientific), followed by cDNA purification with the GeneJET PCR Purification Kit (Thermo Scientific). DNA concentration was determined using a Qubit fluorometer (Thermo Scientific), using the dsDNA High Sensitivity Qubit Assay Kit (Thermo Scientific), as recommended by the manufacturer.

RNA libraries were constructed using the Nextera XT Kit (Illumina, San Diego, CA), and shotgun libraries were prepared using standard Illumina protocols using 1 ng of cDNA. Each pool was indexed according to its provenance using Illumina adaptor-specific indexes, and libraries' fragment size distribution was analyzed using the 2100 Bioanalyzer Instrument (Agilent, Santa Clara, CA). Subsequently, the samples were sequenced on the NextSeq 550 instrument (Illumina) in 150-base paired-end reads.

## Metagenomic analysis

Reads were subjected to quality trimming and filtering using the bbduk program (BBTools v38.42), as described previously (15, 16). Next, *de novo* nucleotide sequence assembly was performed with SPAdes v3.15.3 using the metaSPAdes, metaviralSPAdes, and rnaviralSPAdes parameter options, and MEGAHIT v1.2.9 setting default parameters (17). Assembled contigs longer than 500 nt were clustered using Gclust v1.0 (18), filtered using CheckV v0.8.1 (19), and further analyzed. Viral taxonomic classification of the *de novo*-assembled contigs was performed using Diamond v0.9.14.115 (20) against the NCBI nr protein database and BLASTn. The results of viral taxonomic classification were further summarized to the level of taxonomic families using MEGAN V6.24.0 (21). Contigs related to the family *Coronaviridae* were extracted and compared with coronavirus sequences included in the NCBI database (https://www.ncbi.nlm.nih.gov/) using the Blastn and BlastX algorithms.

## Complete viral genome assembly/scaffolding

Contigs that were related to a unique lowest common ancestor (LCA) according to MEGAN software were aligned and scaffolded. Gaps in the Tb1 genome were filled by sequencing PCR amplicons: five primer pairs were designed using the contig sequence as a template and the Primer3 Plus online tool (https://www.primer3plus.com/amplification; Table S1).

PCRs were performed using a reaction mixture containing 1× PCR buffer, 2.5 mM $MgCl_2$, 0.2 mM dNTPs, 0.8 mM of each primer, and 2.5 U FirePol Taq DNA Polymerase (Solis Biodyne, Tartu, Estonia), with a cycling program of 5 minutes at 95°C, followed by 40 cycles of 30 seconds at 95°C, 30 seconds at 55°C, and 30 seconds at 72°C, with a final extension at 72°C for 5 minutes. PCR amplicons were checked by size determination under UV light after electrophoresis in a 2% agarose gel and ethidium bromide staining. Subsequently, PCR amplicons were purified with spin columns (Nucleospin Gel and PCR Clean-up, Macherey-Nagel) and sequenced using Sanger at a sequencing facility (Joint Laboratory of Aquatic Biotechnology, Faculty of Biochemical and Pharmaceutical Sciences, Argentina).

Coverage statistics of the novel genomes were estimated by remapping the trimmed read data sets to the sequences using Bowtie2 v2.2.6 (22) and by visual inspection with Ugene (v40.0, Unipro) (23).

## Functional annotation

Complete genome sequences of the novel AlphaCoVs were functionally annotated using the online tool Z-curve 2.1 (http://tubic.tju.edu.cn/sars/) (24) and curated manually with the aid of NCBI ORFfinder, SnapGene Viewer 5.0.6 software (Insightful Science, San Diego, CA), and Blastp and Blastn algorithms. Genes and transcription regulatory sequences (TRSs) were located using the CORSID algorithm (25). For each AlphaCoV, the solution with the highest genome coverage was considered optimal.

Putative structural proteins were analyzed and compared against protein databases (nr protein sequences) included in the NCBI using the Blastp algorithm. Sizes, genomic localization, and the 15 expected cleavage sites of the non-structural proteins encoded by ORF1ab were predicted by Z-curve 2.1 (24). Functional domains of putative non-structural proteins were predicted with InterProScan using the integrated protein databases TIGRFAMs, SFLD, PANTHER, HAMAP, PRINTS, Pfam, CATH-Gene3D, ProSiteProfiles, CDD, SUPERFAMILY, and SMART.

## Phylogenetic analysis

The novel genomes and protein-encoding genes (Spike and RdRp) were aligned with selected context sequences of representative species belonging to the family *Coronaviridae* (see Table S2 for accession numbers) using MAFFT v7.453 (26) and ClustalW algorithm v2.1. The RdRp gene data set was constructed with AlphaCoV partial sequences from Argentina ($n = 38$) (27, 28), the U.S. ($n = 143$), and other countries ($n = 21$). Sequences from Beta-, Gamma-, and DeltaCoV genera were included as outgroups ($n = 37$). Phylogenetic analyses were performed using IQtree v1.6.12 (29), and model selection was carried out using the built-in ModelFinder function (26). Branch support was estimated as ultrafast bootstrap support values (30). Pairwise nucleotide similarity plots of complete viral genomes were generated with Simplot v3.5.1 (31), using a window and step size of 1,000 bp and 100 bp, respectively.

## Recombination analysis

Detection of recombinant segments and localization of recombination breakpoints were performed using Recombination Detection Program (RDP) v.4 (32), considering the following methods: RDP (33), GENECONV (34), Bootscan (35), Maxchi (36), Chimaera (37), SiScan (38), and 3Seq (39). Sequences were treated as linear, and the window size for the RDP metric was set at 150 bp; all other parameters were left as default. Recombination events detected by using five or more methods ($P < 0.05$) (40), and those encompassing sequence regions larger than 1,500 bp were considered for further analysis (inclusion criteria). The data set for recombination analysis was compiled, including full-length genomic sequences of the AlphaCoV genus retrieved from GenBank, which were aligned with ClustalW using default parameters. Subsequently, all but one sequence in groups sharing more than 99% nucleotide identity were discarded, yielding a total of 118 sequences in the final data set (Table S3).

## Nucleotide sequence accession numbers

The novel viral genomes and partial sequences reported in this study are available in the GenBank/EMBL/DDBJ database with the following accession numbers: OP715781 (Tadarida brasiliensis bat alphacoronavirus 1 isolate Tb1), OP715780 (Tadarida brasiliensis bat alphacoronavirus 2 isolate Tb2), OP700657 (Tadarida brasiliensis bat alphacoronavirus 2 isolate Tb3), OP729193 (Tadarida brasiliensis bat alphacoronavirus isolate Tb4), OP729194 (Tadarida brasiliensis bat alphacoronavirus isolate Tb5), OP839276 (bat alphacoronavirus isolate Mm1), OP839278 (bat alphacoronavirus isolate Mm2), OP839277 (bat alphacoronavirus isolate Mm3), and OP839279 (bat alphacoronavirus isolate Mm4). The relevant raw high-throughput sequencing data obtained in this study were deposited at the NCBI Sequence Read Archives (SRA) under BioProject ID PRJNA892907.

## RESULTS

### Three novel full-length and partial AlphaCoV genomes were identified in bats of the family Molossidae from Argentina

A total of 47 fecal samples, collected from three species of bats (*T. brasiliensis*, $n = 20$; *E. bonariensis*, $n = 14$; *M. molossus*, $n = 13$) belonging to the family Molossidae, were grouped into six sample pools (Table 1) and included in metagenomic analysis as described previously (15, 16). Fourteen contigs (483 to 28,790 bp long) related to the family *Coronaviridae* were detected in *T. brasiliensis* and *M. molossus* samples (pools 1, 2, and 6), whereas no *Coronaviridae* reads or contigs were identified in samples of *E. bonariensis* (pools 3 and 4; Table 2).

Three novel full-length AlphaCoV genomes were identified in *T. brasiliensis* samples, with two contigs being assigned to the same AlphaCoV LCAs (MZ081383, KY799179, and NC022103), which corresponded to complete novel genomes of two viruses, Tadarida brasiliensis bat alphacoronavirus 2 isolate Tb2 (28,719 bp) and Tb3 (28,790 bp), identified

**TABLE 2** Contig characteristics mapped to the family *Coronaviridae* in each sample pool[a]

| Pool | Bat host | Coronavirus contig ID | Length (bp) | Average coverage depth | Genome regions | Virus name | Collection year | Isolate name | Genbank accession number |
|---|---|---|---|---|---|---|---|---|---|
| 1 | *Tadarida brasiliensis* | k149_1310_3 flag = 1 multi = 11.0000 len = 11528 | 11,528 | 10.26 | ORF1ab, partial; S, partial | *Tadarida brasiliensis bat alphacoronavirus 1* | 2017 | Tb1 | OP715781 |
| | | k149_1373 flag = 1 multi = 14.0000 len = 5133 | 5,133 | 11.93 | ORF1a, partial | | | | |
| | | k149_656 flag = 1 multi = 23.0000 len = 3686 | 3,686 | 18.27 | S, partial; ORF3, partial | | | | |
| | | k149_1151 flag = 1 multi = 29.0000 len = 3026 | 3,026 | 23.39 | E, M, N, ORF7 | | | | |
| | | k149_1155 flag = 1 multi = 11.0000 len = 3009 | 3,009 | 9.41 | ORF1a, partial | | | | |
| | | NODE_73_length_2349_cov_2.540000 | 2,349 | 8.50 | ORF1a, partial | | | | |
| | | k149_1770_3 flag = 1 multi = 491.6667 len = 28719 | 28,719 | 371.53 | Complete genome | *Tadarida brasiliensis bat alphacoronavirus 2* | 2017 | Tb3 | OP700657 |
| | | NODE_11_length_6460_cov_13.837290 | 646 | 132.35 | ORF1ab (Nsp16), partial; S, ORF3, E and M complete genes | *Tadarida brasiliensis bat alphacoronavirus* | 2017 | Tb4 | OP729193 |
| | | k149_1346_flag_0_multi_129.0000_len_5703 | 5,703 | 176.01 | ORF1ab (Nsp16), partial; S and ORF3 complete genes | *Tadarida brasiliensis bat alphacoronavirus* | 2017 | Tb5 | OP729194 |
| 2 | *Tadarida brasiliensis* | k149 2895 two flag = 1 multi = 892.0000 len = 28811 | 28,790 | 676.85 | Complete genome | *Tadarida brasiliensis bat alphacoronavirus 2* | 2016 | Tb2 | OP715780 |
| 3 | *Eumops bonariensis* | Not detected | | | | | | | |
| 4 | *Eumops bonariensis* | Not detected | | | | | | | |
| 5 | *Molossus molossus* | Not detected | | | | | | | |
| 6 | *Molossus molossus* | NODE 148 length 1094 cov 1.167839 | 1,094 | 3.68 | ORF1ab, partial | *Bat alphacoronavirus* | 2017 | Mm1 | OP839276 |
| | | k149 1587 flag = 1 multi = 2.0000 len = 599 | 597 | 2.74 | ORF1ab, partial | *Bat alphacoronavirus* | 2017 | Mm2 | OP839278 |
| | | k149 101 flag = 1 multi = 3.0000 len = 532 | 532 | 4.20 | S gene, partial | *Bat alphacoronavirus* | 2017 | Mm3 | OP839277 |
| | | k149 1871 flag = 1 multi = 3.0000 len = 483 | 483 | 3.22 | ORF1ab, partial | *Bat alphacoronavirus* | 2017 | Mm4 | OP839279 |

[a]Novel AlphaCoV sequences identified in this work are given in italics.

in Pools 1 and 2, respectively (Table 2). In addition, Pool 1 contained several additional contigs, which were assigned to another unique AlphaCoV LCA (MW924112; strain HCQD2020). These contigs were scaffolded by alignment to the LCA, and the gap regions were completed with Sanger sequences of PCR amplicons (five amplicons; sizes ranging from 501 to 600 bp). Finally, the complete genome sequence Tb1 (28,844 bp) of a novel AlphaCoV was obtained (Tadarida brasiliensis bat alphacoronavirus 1 isolate Tb1). The novel AlphaCoVs exhibited G + C contents of 40.8%, 43.2%, and 43.3% for Tb1, Tb2, and Tb3, respectively.

Finally, an additional six partial sequences that corresponded to structural and non-structural genes of AlphaCoV (Table 2; Fig. 1) were found, with two contigs approximately 6,000 bp long related to LCA MK472070 identified in *T. brasiliensis* (Pool 1) and four contigs ranging from 483 to 1,094 bp related to LCAs NC022103 and MW924112 detected in *M. molossus* (Pool 6).

## The novel AlphaCoVs Tb1, Tb2, and Tb3 encode all five characteristic coronavirus open reading frames (ORFs)

Sequence annotation and BLASTP analysis showed that all novel viruses encoded the five characteristic ORFs found in *Coronaviridae* family, including replicase polyprotein Orf1ab, Spike glycoprotein (S), envelope (E), membrane (M), and nucleocapsid (N) proteins (Fig. 1). TRS locations were predicted to confirm gene locations (Table 3). The "TTTAAAC"

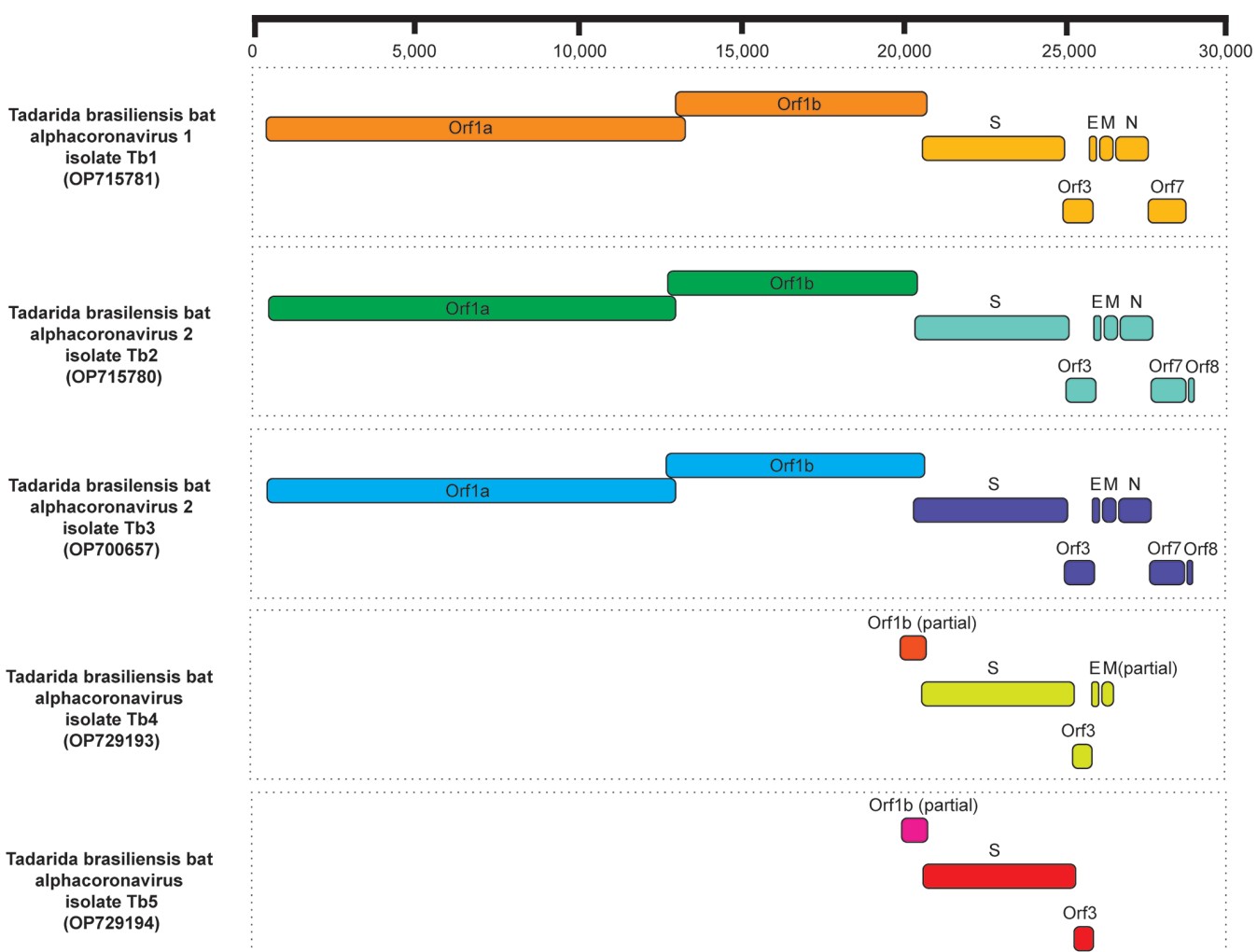

**FIG 1** Genome organization of novel Alphacoronavirus genomes and sequences identified in *T. brasiliensis*. S, Spike; E, envelope; M, membrane; N, nucleocapsid.

**TABLE 3** Putative ORFs and TRS positions of the novel AlphaCoVs[a]

| Putative ORFs | Tb1 | | | Tb2 | | | Tb3 | | |
|---|---|---|---|---|---|---|---|---|---|
| | Length (nt/aa) | TRS location | TRS sequence(s) (Distance to ATG) | Length (nt/aa) | TRS location | TRS sequence(s) (Distance to ATG) | Length (nt/aa) | TRS location | TRS sequence(s) (Distance to ATG) |
| ORF1ab | 20,595/6,864 | 83 | TCAACTAAACGA(218)ATG | 20,447/6,815 | 76 | TCTCAACTAAAC(217)ATG | 20,447/6,815 | 78 | TCTCAACTAAAC(217)ATG |
| S | 4,044/1,347 | 20.895 | TCAACCAAATG | 4,305/1,434 | 20.738 | ATTCAACTAAATAAAACTATG | 4,344/1,447 | 2.074 | ATTCAACTAAATAAAATG |
| ORF3 | 675/224 | 24.896 | TCAACTAAACT (36)ATG | 684/227 | 25.016 | CATCAACTAAAC (37)ATG | 684/227 | 25.054 | CATCAACTAAAAC (37)ATG |
| E | 228/75 | 25.593 | AAAACTTTACGAAGATG | 228/75 | 25.714 | GTTCAACTTGACGAATATG | 228/75 | 25.752 | GTTCAACTTGACGAATATG |
| M | 678/225 | 25.831 | CTAACTAAATCAAAATG | 771/256 | 25.953 | TCTAAACGAAAATG | 768/255 | 25.991 | TCTAAACGAAATG |
| N | 1,125/374 | 26.523 | TCAATTAAACA (6)ATG | 1,308/435 | 26.729 | TCTAAACTAAACAAAATG | 1,308/435 | 26.764 | TCTAAACTAAACAAAATG |
| ORF7 | 828/275 | 27.657 | CCAACTAAACATG | 381/126 | 28.050 | AATCAACTAAAACATG | 381/126 | 28.085 | AATCAACTAAAACATG |
| ORF8 | - | - | - | 144/47 | 28.335 | TCCACAACCACCT(98)ATG | 144/47 | 28.370 | TCCACAACCACCT(98)ATG |

[a]TRS: transcription regulatory sequences. TRS: core sequences are underlined.

conservative heptameric sequence between Orf1a and Orf1b, required for −1 ribosomal frameshift Orf1b expression (41), was present in all three novel AlphaCoVs. Two putative accessory genes (ORF3 located between S and E, and ORF7 located downstream from N gene) were also identified in all novel AlphaCoVs (Table 3). Blastn and BlastX searches showed low similarities of the ORF7 sequences of Tb2 and Tb3 to the most closely related database proteins (34% identity with NS7 protein Megaderma bat coronavirus; URD31312.1), and an additional putative ORF8 was recognized in Tb2 and Tb3, which also did not retrieve homological sequences in the databases. Furthermore, analysis of the replicase polyprotein (Orf1ab) showed the typical 16 putative non-structural proteins (Nsp1–Nsp16) with their corresponding cleavage sites (Table 4).

## Phylogenetic analysis shows two AlphaCoV lineages simultaneously circulating in the colony of *T. brasiliensis*

All three novel AlphaCoVs phylogenetically positioned outside the 15 subgenera currently recognized by the International Committee on Taxonomy of Viruses (ICTV; Fig. 2A). In addition, none of them were positioned alongside the AlphaCoV identified in a *Chaerephon plicatus* (Buchannan, 1800) bat, and so far, the only AlphaCoV that has been reported in molossids. In fact, Tb1 gravitated more toward viruses identified in bats of the genus *Eptesicus* from the U.S. and South Korea. This group appeared to be related to the AlphaCoV subgenus *Myotacovirus* and viruses identified in bats of the genus *Myotis* from China and Denmark (Fig. 2A). On the other hand, Tb2 and Tb3 appeared to be variants of the same virus, forming a novel monophyletic clade, close to the subgenus *Colacovirus*, sharing between 98% and 100% amino acid (aa) and 97% nt similarity along the genome, with the exception of the Spike gene (specifically, along the S1 domain), with an aa similarity of 91.3% (85.8% in nt; Fig. S1). Interestingly, Tb2 and Tb3 were found in the same bat colony, in pools from different collection seasons: Pool 1 in 2016 and Pool 2 in 2017 (Table 2).

Pairwise aa comparisons of the replicase polyprotein pp1ab conserved domains (Table S4) showed 95.8% aa identity of Tb1 with a bat coronavirus (OL415262) identified in *Eptesicus fuscus* (Beauvois, 1796) from the U.S. and 84.2% aa identity for both Tb2 and Tb3 with the prototype of the subgenus *Decacovirus* identified in *Rhinolophus ferrumequinum* (Schreber, 1774) in China (NC028814). Considering the current criterion established by the ICTV for species demarcation (42) (sequences with <92.5% aa identity in this region with respect to other known AlphaCoV isolates, https://ictv.global/report/chapter/coronaviridae/coronaviridae, accessed on 14 March 2023), it is possible that Tb1 is a member of the same species of *Eptesicus* coronaviruses, whereas Tb2 and Tb3 could be considered prototypes of a novel species within the genus AlphaCoV.

To explore phylogenetic associations in the Spike protein, Tb4 and Tb5 partial sequences were also included in the data set (Fig. 2; Fig. S2). Although *T. brasiliensis* genomes clustered with the same viruses as the full-length genome tree (Fig. 2A), their

**TABLE 4** Putative non-structural proteins (NSPs) and cleavage sites of polyproteins 1a and 1ab of the novel AlphaCoVs[a]

| Putative Nsp | Tb1 First-last amino acid residues | Protein size | Cleavage sequence | Tb2 First-last amino acid residues | Protein size | Cleavage sequence | Tb3 First-last amino acid residues | Protein size | Cleavage sequence | Putative protein function and functional domains (InterPro entry accession) |
|---|---|---|---|---|---|---|---|---|---|---|
| Nsp 1 | 311–641 | 110 | FGHCGG\|TPCVNT | 303–632 | 110 | FGRRGG\|NVVYVD | 305–634 | 110 | FGRRGG\|NVVYVD | Host gene expression supression. IPR046443 |
| Nsp 2 | 642–2974 | 778 | FTFKRG\|GGVTFG | 633–2975 | 781 | YRKKGG\|GGVAFA | 635–2,977 | 781 | YRKKGG\|GGVAFA | Unknown function. IPR044385 |
| Nsp 3 | 2,975–8,089 | 1,705 | IVQKSG\|SGPFP | 2,976–7,964 | 1663 | ANKKGA\|GELREC | 2,978–7,966 | 1,663 | ANKKGA\|GELREC | Papain-like protease. IPR013016 (papain-like protease), IPR044357 (ubiquitin-like domain 1), IPR002589 (Macro domain), IPR044353 (ubiquitin-like domain 2), IPR043611 (C-terminal domain) |
| Nsp 4 | 8,090–9,523 | 478 | STLQ\|AGLR | 7,965–9,398 | 478 | STLQ\|SGLR | 7,967–9,400 | 478 | STLQ\|SGLR | Replication-transcription complex formation. IPR043612 (N-terminal domain), IPR032505 (C-terminal domain) |
| Nsp 5 | 9,524–10,429 | 302 | VTLQ\|SGRK | 9,399–10,304 | 302 | VTLQ\|SGKT | 9,401–10,306 | 302 | VTLQ\|SGKT | 3C-like proteinase. IPR008740 |
| Nsp 6 | 10,430–11,266 | 279 | SSVQ\|SKLT | 10,305–11,135 | 277 | STVQ\|SKLT | 10,307–11,137 | 277 | STVQ\|SKLT | Double-membrane vesicles induction. IPR044369 |
| Nsp 7 | 11,267–11,515 | 83 | AMLQ\|SIAS | 11,136–11,384 | 83 | TILQ\|SVAA | 11,138–11,386 | 83 | TILQ\|SVAA | Viral RNA replication complex. IPR014828 |
| Nsp 8 | 11,516–12,100 | 195 | VKLQ\|NNEI | 11,385–11,969 | 195 | VKLQ\|NNEI | 11,387–11,971 | 195 | VKLQ\|NNEI | Viral RNA replication complex. IPR014829 |
| Nsp 9 | 12,101–12,427 | 109 | IRLQ\|AGKQ | 11,970–12,293 | 108 | VRLQ\|AGKQ | 11,972–12,295 | 108 | VRLQ\|AGKQ | Single-stranded RNA-binding viral protein. IPR014822 |
| Nsp 10 | 12,428–12,832 | 135 | ANVQ\|SFDQ | 12,294–12,698 | 135 | TVMQ\|SLDT | 12,296–12,700 | 135 | TVMQ\|SLDT | Involved in RNA synthesis. IPR018995 |
| Nsp 11 | 12,833–12,886 | 18 | - | 12,699–12,752 | 18 | - | 12,701–12,754 | 18 | - | Short peptide at the end of Orf1a |
| Nsp 12 | 12,833–15,612 | 927 | TVLQ\|ASGM | 12,699–15,478 | 927 | TVLQ\|AAGL | 12,701–15,480 | 927 | TVLQ\|AAGL | RNA-dependent RNA polymerase. IPR044356 |
| Nsp 13 | 15,613–17,403 | 597 | ADLQ\|ATDG | 15,479–17,269 | 597 | SDLQ\|SNGD | 15,481–17,271 | 597 | SDLQ\|SNGD | Helicase. IPR044343 (1B domain), IPR027351 (helicase core domain), IPR027352 (ZBD domain) |
| Nsp 14 | 17,404–18,960 | 519 | TRMQ\|GLEN | 17,270–18,826 | 519 | VNLQ\|GLEN | 17,272–18,828 | 519 | VNLQ\|GLEN | Exoribonuclease and Guanine-N7 methyl transferase. IPR009466 |
| Nsp 15 | 18,961–20,001 | 347 | PQLQ\|SAEW | 18,827–19,843 | 339 | PQLQ\|SSEW | 18,829–19,845 | 339 | PQLQ\|SSEW | Endoribonuclease. IPR043606 (N-terminal oligomerization domain), IPR044322 (non-catalytic middle domain), IPR043609 (C-terminal NendoU catalytic domain) |
| Nsp 16 | 20,002–20,901 | 300 | - | 19,844–20,746 | 301 | - | 19,846–20,748 | 301 | - | O-methyltransferase. IPR009461 |

[a]Representative functional domains identified by InterProScan are shown by InterPro accession numbers with its reported function in the databases.

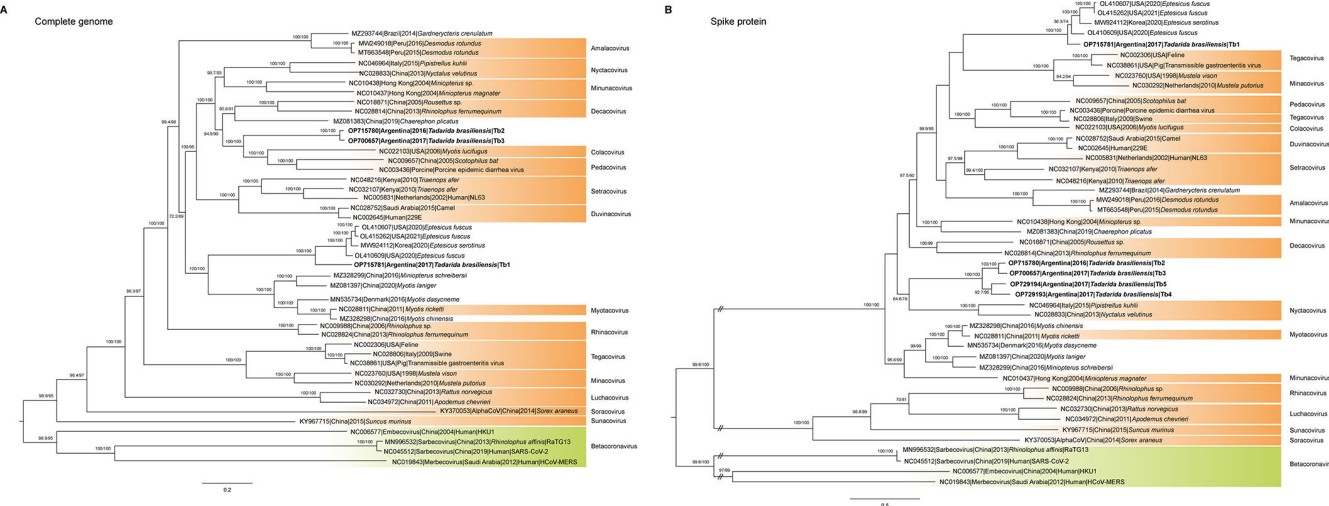

**FIG 2** Phylogenetic analysis of novel AlphaCoVs identified in this work and representative viruses from different subgenera. Phylogenetic trees of (A) the full-length virus genomes (nt) and (B) the spike protein of AlphaCoVs (aa). For the phylogenetic tree of the Spike gene (nt), see Fig. S2. The phylogenetic trees were constructed using IQtree v1.6.12 (29) with 1,000 UFBootstrap (30) and SH-aLRT branch test replicates. The phylogenetic model GTR was chosen as the best-fitting model. All trees were rooted using BetaCoVs as outgroups. Tree visualization was facilitated using Figtree v1.4.4 (https://github.com/rambaut/fig-tree.git, accessed on 25 September 2022). AlphaCoV subgenera (orange) and BetaCoVs (green) are shown. Newly identified AlphaCoVs are given in bold. Node bootstrap support values <60 are not shown. Branch lengths are scaled according to the number of substitutions per site.

associations with the AlphaCoV subgenera changed, suggesting different evolutionary forces shaping the Spike gene phylogeny. In particular, Tb1 was related to members of the subgenera *Tegacovirus* and *Minacovirus*, which include AlphaCoVs identified in animals different from bats (Fig. 2B). On the other hand, Tb4 and Tb5 grouped together with Tb2 and Tb3 into a monophyletic clade, close to the genus *Nyctacovirus*. The changes observed in the Spike gene phylogenetic tree topology (Fig. 2B; Fig. S2) suggest that the novel viruses could be involved in recombination events and deserve further analysis.

Finally, the phylogenetic relationships of partial RdRp genes were investigated in order to explore the associations of the novel viruses with the recently described AlphaCoV lineages circulating in bats from the Americas (43). As shown in Fig. 3, Tb1 clustered into clade A, together with other AlphaCoV sequences identified in *T. brasiliensis* individuals from Argentina, whereas Tb2 and Tb3 variants grouped into clade B, closely related to strains previously reported in *T. brasiliensis*, *Myotis* spp., and *Molossus* spp. individuals from Argentina.

Altogether, these findings indicate that two AlphaCoV lineages are circulating in the *T. brasiliensis* colony, with the Tb2/Tb3 lineage possibly involved in persistent infections.

## Local sequence context convergence in the Spike gene suggests proxied recombinant transfer between Tb1 and an *Eptesicus* bat AlphaCoV

To investigate whether the novel viral genomes identified herein belong to recombinant AlphaCoV lineages, they were analyzed with selected AlphaCoV context sequences using RDP software. Overall, 97 statistically significant events were detected using five or more methods, of which 38 met the inclusion criteria (recombinant region length >1,500 bp; Table S5). A single recombination event that involved the Spike gene in one of the novel genomes (Tb1) was found with six out of seven methods (*P*-values ranging from $1.31 \times 10^{-51}$ to $2.54 \times 10^{-3}$; Fig. 4, upper panel; Table S5). There was a local sequence convergence around the Spike gene between the *Eptesicus* bat coronavirus strain 16964 (OL410609; recognized as the putative recombinant) detected in South Dakota in 2020, globally most closely related to the *Eptesicus* bat coronavirus strain 15712 (OL410607; recognized as the putative major parent), and the novel isolate

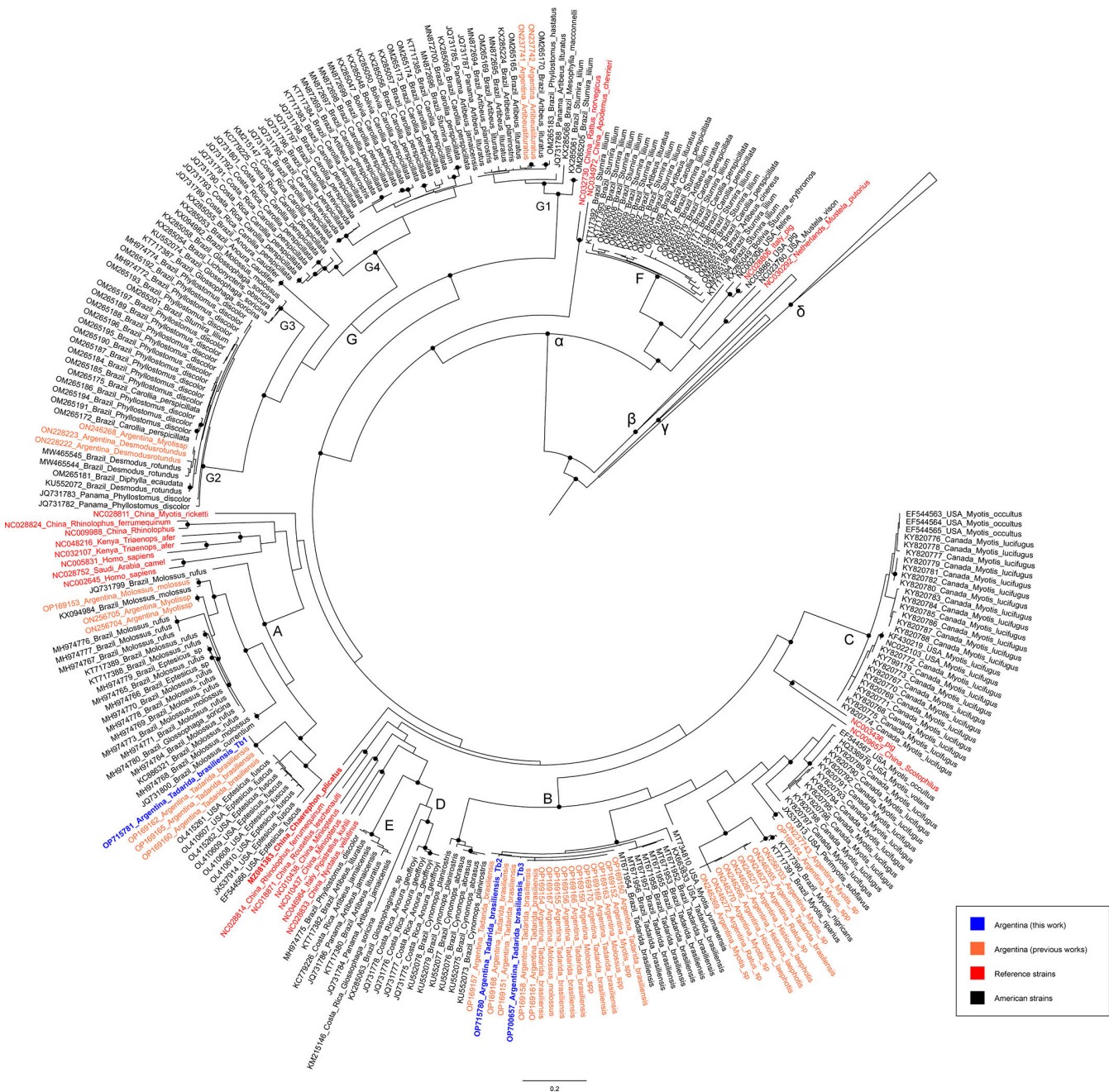

**FIG 3** Phylogenetic tree by maximum likelihood based on partial RdRp gene sequences from the Americas and worldwide. Sequences identified in this work are given in blue. The data set included sequences from the Americas (black), sequences from other reports from Argentina (orange), and reference AlphaCoVs (red). GammaCoV and DeltaCoV genera were used as outgroups and are shown collapsed. Clades A–G previously defined by others are shown (43). Nodes with bootstrap support values > 99 are represented with black dots. Branch lengths are scaled according to the number of substitutions per site.

Tb1 (OP715781; recognized as the putative minor parent; Fig. 4, bottom panel). The recombination hypothesis was further reinforced with local tree incongruence analysis (Fig. 5).

The recombinant segment extended between 21,796 and 23,414 bp in the recombinant sequence and represented a 1,618-nt fragment within the Spike coding region, encompassing part of the C-terminal S1 and the N-terminal S2 subdomains. Nucleotide sequence similarities between the genomes involved in the recombination event ranged from 79.8% to 93.9% at the full-genome level (data not shown). Of note, the relatively

| Method | RDP | Bootscan | Maxchi | Chimaera | SiScan | 3Seq |
|---|---|---|---|---|---|---|
| p-value | $1.31 \times 10^{-51}$ | $1.92 \times 10^{-34}$ | $1.44 \times 10^{-8}$ | $1.24 \times 10^{-17}$ | $2.54 \times 10^{-3}$ | $4.15 \times 10^{-10}$ |

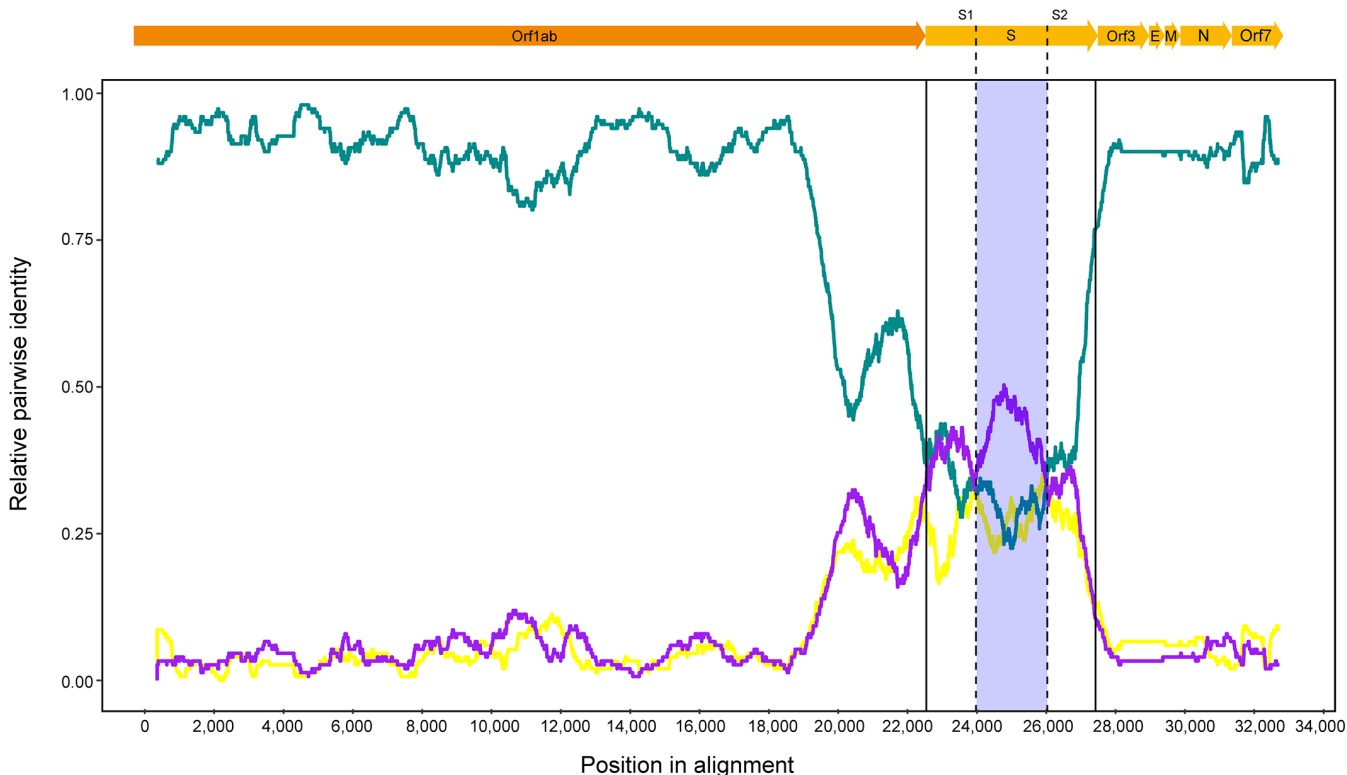

**FIG 4** Characterization of a recombination event involving the novel AlphaCoV Tb1. The recombination methods that provided statistically significant *P*-values are indicated (upper panel). Relative pairwise identities between the recombinant and minor parent (purple), the recombinant and major parent (blue), and both parents (yellow) are shown (bottom panel). The region between the recombination breakpoints is shaded and limited by dashed lines in the Tb1 genomic map. Limits of the Spike gene and S1/S2 subdomains are also shown.

low mid-range pairwise nt similarity between Tb1 and OL410609 at the recombinant segment (79%, data not shown) could be consistent with a proxied recombinant transfer scenario.

## DISCUSSION

To date, coronavirus studies carried out on bats from Argentina have corresponded to surveillance campaigns based on the detection and sequencing of a conserved RdRp region (27, 28). This strategy has been useful in exploring the presence, diversity, and phylogenetic relationships of coronaviruses circulating in some species of bats. However, these approaches do not reflect the full evolutionary history and diversity of these viruses and do not allow the study of recombination as a mechanism shaping the phylogeny of the family *Coronaviridae*. In addition, highly divergent coronaviruses might remain undetected because the majority of bat-CoV sequences that are available in public databases have been generated with directed primers (44).

Even though AlphaCoV and BetaCoV genera have been detected in bats worldwide, AlphaCoVs might be more diverse, common, and widespread (3). In fact, more than 80% of coronavirus sequences identified in American bats belong to the genus AlphaCoV (45). In this study, using a viral particle enrichment strategy and viral (meta)genome shotgun sequencing, we found evidence of AlphaCoV infection in the feces of two out of the three species of bats of the family Molossidae investigated: *T. brasiliensis* and *M. molossus*. We identified and characterized three novel AlphaCoVs from the colony of *T. brasiliensis* at Rosario, which, to our knowledge, are the first full-length genomes

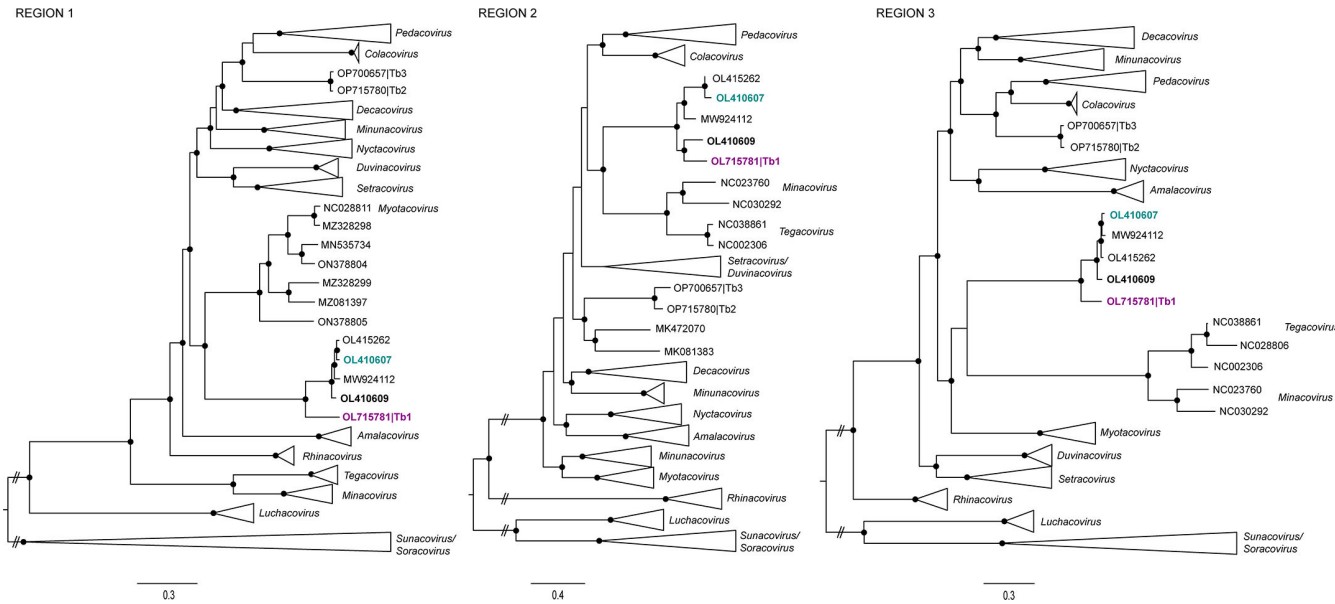

**FIG 5** Phylogenetic tree incongruence analysis. Maximum likelihood trees from REGION 1 (nt positions 1–21,795), REGION 2 (nt positions 21,796–23,414), and REGION 3 (nt positions 23,415–28,340). The recombinant strain is indicated in bold; the major and minor parents are indicated in blue and purple, respectively. For clarity, AlphaCoV subgenera and related sequences have been collapsed. Nodes with bootstrap support values >90 are represented with black dots.

reported in this family of bats in the Americas. We also provided the first evidence of AlphaCoV infection in *M. molossus* individuals inhabiting the province of Santa Fe, but the sequences recovered were too short to accurately and comprehensively evaluate their phylogenetic relationships. Therefore, wider sampling efforts are required to obtain full genomes of viruses infecting this species of bats and to increase knowledge of the coronavirus diversity of the family Molossidae in the Americas.

Isolates Tb1, Tb2, and Tb3 exhibited the typical genome organization found in the genus AlphaCoV, with five common ORFs, 16 typical putative non-structural proteins of the Orf1ab polyprotein (42, 46, 47), and accessory proteins. The phylogenomic analysis showed that all novel AlphaCoVs fell outside the current recognized subgenera, with isolates Tb2 and Tb3 most probably constituting novel putative species according to the ICTV demarcation criteria (42). In addition, the novel AlphaCoVs were positioned into two different lineages: the Tb1 lineage, which was closely related to viruses identified in bats of the genus *Eptesicus* from the U.S. and South Korea, and the Tb2/Tb3 lineage, which grouped together with *Myotis* AlphaCoVs from the U.S. This observation was confirmed by the RdRp phylogenetic tree because the Tb1 isolate grouped into the previously proposed clade A, whereas Tb2 and Tb3 isolates clustered together in clade B with sequences identified in molossid bats from Argentina (43). In contrast, none of the novel viruses were closely related to the AlphaCoV identified in a molossid bat from China (5), suggesting that coronavirus circulation might probably be related to geographical distribution and virus-host co-divergence, as reported previously (27, 48). Indeed, it has been proposed that *E. fuscus* (U.S.) and *E. serotinus* (South Korea) may be conspecific (49), implying a single *Eptesicus* AlphaCoV species (10). Interestingly, isolate Tb1 is also highly similar to *Eptesicus* AlphaCoV sequences, suggesting a possible host jump event between both bat species. In line with this, previous studies proposed that AlphaCoV cross-species transmission might have occurred between Molossidae and Vespertilionidae bat families in the past (43).

Cross-species transmission has been considered a common evolutionary force during coronavirus evolution, with the events mentioned more likely occurring between sympatric bat hosts (48). In fact, roost sharing among individuals of *T. brasiliensis* and *E. fuscus* has been reported at the Desert Museum of Tucson, Arizona, in Ensenada

(Baja California, Mexico) (50) and in caves of Jalisco (Mexico) (51). Moreover, habitat sharing has been proposed as a key factor that increases the likelihood of viral sharing among species of bats (52) and is a prerequisite for recombination(s) to occur. In line with this observation, a putative recombination event has been detected in the Spike gene, involving the novel Tb1 and two *E. fuscus* AlphaCoVs (OL410609 and OL410607), identified in South Dakota in 2020.

The levels of global (along the complete genome) and local (at the recombinant segment) pairwise sequence similarities suggested a scenario involving proxied recombinant transfer in which the transferred sequence context is part of a larger gene pool that could span viruses infecting co-roosting bats. On the other hand, although no recombination events involving Tb2 and Tb3 isolates were detected in this data set analysis, the dissimilarity of their Spike genes suggested a past recombination event of unknown origin. Nevertheless, it is possible that the parental viruses have not been identified yet, and the possibility of cumulative nucleotide substitutions in the Spike gene, due to their likely involvement in persistent infections, cannot be excluded, as observed in previous studies of AlphaCoV persistence in natural (53, 54) and artificial populations of *Myotis* (9, 55).

Interestingly, the putative recombinant regions—one involving Tb1 detected by RDP and hypothetical Tb2/Tb3—spanned the S1 subdomain of the Spike gene, which encompassed the potential receptor-binding domain (RBD) of AlphaCoVs. The transfer of small sections of Spike protein during recombination in a coinfection, particularly involving the RBD domain, may be beneficial for the virus to broaden its host range and in intra-host immune suppression (56). Moreover, phylogenetic analysis of the Spike protein showed changes in the topology with respect to the full-length genome tree, in agreement with previous observations (5, 57). In fact, this split in the phylogenetic history of the Spike proteins is characteristic of the family *Coronaviridae* (42), suggesting that the Spike gene and the rest of the genome have distinct evolutionary patterns. This semi-independent evolution of structural and non-structural genes might be a common strategy of RNA viruses (58, 59).

Because susceptible newborn bats would amplify the virus due to their immature immune system, infecting adult females, it has been proposed that maternal colonies play an important role in maintaining coronavirus infection at the population level (16, 60, 61). In contrast, non-gregarious bats would experience mostly self-limited infections (54, 61). Of note, the novel AlphaCoVs identified in the maternal colony of *T. brasiliensis* had high sequence depth, suggesting that the animals may be shedding the virus, as was observed previously with a circovirus infection (16). In light of these observations, further efforts are needed to elucidate the transmission dynamics of bat-borne viruses in South America, underscoring longitudinal studies to understand their maintenance patterns and zoonotic potential.

## Conclusions

This study generated three novel complete AlphaCoV genomes identified in individuals of *T. brasiliensis*, which showed two different evolutionary patterns and are the first to be reported in the family Molossidae in the Americas. In addition, the Tb1 isolate was involved in a putative recombination event with AlphaCoVs identified in bats of the genus *Eptesicus* from the U.S., whereas Tb2 and Tb3 isolates were found in different collection seasons and might be involved in persistent viral infections in the bat colony. These findings contribute to the knowledge of the global diversity of bat coronaviruses in poorly studied species and highlight the different evolutionary aspects of AlphaCoVs circulating in bat populations in Argentina. Greater efforts are needed for long-term surveillance and full-genome characterization to better understand coronavirus evolution in bat populations in the Americas and to elucidate the mechanisms involved in facilitating cross-species transmission.

## ACKNOWLEDGMENTS

The authors thank Irene Villa, German Saigo, Mauricio Taborda, and Valeria Olivera for collecting and processing the bat samples.

This research was funded by (i) the National Agency for the Promotion of Science and Technology (PICT-2019–01790; PICT-2020–00571); (ii) the European Society of Clinical Microbiology and Infectious Diseases (ESCMID) Observership Program, granted to Agustina Cerri (ESCMID Observership no. 1971); and (iii) the Slovenian Research Agency, grant no. P3-00083. Agustina Cerri was supported by doctoral fellowships from CONICET.

A.C.: conceptualization, investigation, methodology, and writing—original draft preparation; E.M.B.: conceptualization, formal analysis, resources, methodology, writing—original draft preparation, and review and editing; T.M.Z.: conceptualization, methodology, formal analysis, visualization, and writing—original draft preparation; A.R.: conceptualization, resources, and writing—review and editing; M.E.M.: resources and writing—review and editing; L.H.: writing—review and editing; P.E.C.: conceptualization, visualization, and writing—review and editing; V.D.D.: resources; R.M.B.: resources and writing—review and editing; M.P.: conceptualization, supervision, resources, and writing—review and editing; and A.A.G.: conceptualization, supervision, resources, and writing—review and editing. All authors have read and agreed to the published version of the manuscript.

The authors declare no conflict of interest. The funders had no role in the design of the study; collection, analyses, or interpretation of data; writing of the manuscript; or the decision to publish the results.

## AUTHOR AFFILIATIONS

[1]Human Virology Group, Rosario Institute of Molecular and Cellular Biology (IBR-CONICET), Rosario, Argentina

[2]Virology Area, Faculty of Biochemical and Pharmaceutical Sciences, National University of Rosario, Rosario, Argentina

[3]Bat Conservation Program of Argentina, San Miguel de Tucumán, Argentina

[4]Institute of Microbiology and Immunology, Faculty of Medicine, University of Ljubljana, Ljubljana, Slovenia

[5]Dr. Ángel Gallardo Provincial Museum of Natural Sciences, Rosario, Argentina

[6]Argentine Biodiversity Research Institute (PIDBA), Faculty of Natural Sciences, National University of Tucumán, San Miguel de Tucumán, Argentina

[7]Institute of Virology and Technological Innovations (INTA/CONICET), Castelar, Argentina

[8]Robert Koch Institute, Berlin, Germany

[9]DETx MOL S.A. La Segunda Núcleo Corporate Building, Alvear, Argentina

## AUTHOR ORCIDs

Elisa M. Bolatti  http://orcid.org/0000-0001-6467-0650
Adriana A. Giri  http://orcid.org/0000-0003-4925-9075

## FUNDING

| Funder | Grant(s) | Author(s) |
| --- | --- | --- |
| Agencia Nacional de Promoción de la Investigación, el Desarrollo Tecnológico y la Innovación (Agencia I+D+i) | PICT-2019-01790 | Elisa M. Bolatti |
| Agencia Nacional de Promoción de la Investigación, el Desarrollo Tecnológico y la Innovación (Agencia I+D+i) | PICT-2020-00571 | Adriana A. Giri |
| European Society of Clinical Microbiology and Infectious Diseases (ESCMID) | Observership no. 1971 | Agustina Cerri |

| Funder | Grant(s) | Author(s) |
|---|---|---|
| Slovenska Akademija Znanosti in Umetnosti (SAZU) | P3-00083 | Mario Poljak |
| Consejo Nacional de Investigaciones Científicas y Técnicas (CONICET) | Ph.D fellow | Agustina Cerri |
| Consejo Nacional de Investigaciones Científicas y Técnicas (CONICET) | Ph.D fellow | Violeta Di Domenica |

## AUTHOR CONTRIBUTIONS

Agustina Cerri, Conceptualization, Investigation, Methodology, Writing – original draft | Elisa M. Bolatti, Conceptualization, Formal analysis, Methodology, Resources, Writing – original draft, Writing – review and editing | Tomaz M. Zorec, Conceptualization, Formal analysis, Methodology, Visualization, Writing – original draft | Maria E. Montani, Resources, Writing – review and editing | Agustina Rimondi, Conceptualization, Resources, Writing – review and editing | Lea Hosnjak, Writing – review and editing | Pablo E. Casal, Conceptualization, Visualization, Writing – review and editing | Violeta Di Domenica, Resources | Ruben M. Barquez, Resources, Writing – review and editing | Mario Poljak, Conceptualization, Resources, Supervision, Writing – review and editing | Adriana A. Giri, Conceptualization, Resources, Supervision, Writing – review and editing

## DATA AVAILABILITY

The sequences of novel viruses reported in this article are openly available in the GenBank/EMBL/DDBJ database with the following accession numbers: OP715780–OP715781, OP700657, OP729193–OP729194, and OP839276–OP839279. The relevant raw high-throughput sequencing data obtained in this study were deposited at the NCBI Sequence Read Archives (SRA) under BioProject ID PRJNA892907.

## ETHICS APPROVAL

The animal study protocol was approved by the Ministry of Environment of the Argentinian Province of Santa Fe (Files 519/17 and 356) and the Animal Ethics Committee of the Faculty of Pharmaceutical and Biochemical Sciences (National University of Rosario, Rosario, Argentina, File 6060/243).

## ADDITIONAL FILES

The following material is available online.

### Supplemental Material

**Fig. S1 (Spectrum02047-23-S0001.tif).** Sequence nucleotide similarity between Tb2 and Tb3.
**Fig. S2 (Spectrum02047-23-S0002.tif).** Phylogenetic tree of the Spike gene (nt).
**Table S1 (Spectrum02047-23-S0003.docx).** Sequences of primers designed using Primer3 Plus tool.
**Table S2 (Spectrum02047-23-S0004.docx).** CoV sequences used for phylogenetic analysis.
**Table S3 (Spectrum02047-23-S0005.docx).** AlphaCoVs sequences used in the recombination analysis.
**Table S4 (Spectrum02047-23-S0006.docx).** Pairwise amino acid comparisons of the replicase polyprotein pp1ab conserved domains of the AlphaCoVs sequences.
**Table S5 (Spectrum02047-23-S0007.docx).** Recombination events meeting the inclusion criterion.

Open Peer Review

**PEER REVIEW HISTORY (review-history.pdf).** An accounting of the reviewer comments and feedback.

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
