## [Reviewer comments · Microbiology Spectrum]

Microbiology Spectrum

Identification and Characterization of Novel Alphacoronaviruses in *Tadarida brasiliensis* (Chiroptera, Molossidae) from Argentina: Insights into Recombination as a Mechanism Favoring Bat Coronavirus Cross-Species Transmission

Agustina Cerri, Elisa Bolatti, Tomaž Mark Zorec, María Montani, Agustina Rimondi, Lea Hošnjak, Pablo Casal, Violeta Di Domenica, Ruben Barquez, Mario Poljak, and Adriana Giri

Corresponding Author(s): Adriana Giri, Human Virology Group, National Council of Scientific and Technical Research, Institute of Molecular and Cell Biology of Rosario, Rosario, Argentina

Review Timeline:

Submission Date:	May 16, 2023
Editorial Decision:	June 23, 2023
Revision Received:	July 13, 2023
Accepted:	July 14, 2023

Editor: Biao He

Reviewer(s): The reviewers have opted to remain anonymous.

Transaction Report:

DOI: <https://doi.org/10.1128/spectrum.02047-23>

June 23, 2023

Prof. Adriana A Giri
Human Virology Group, National Council of Scientific and Technical Research, Institute of Molecular and Cell Biology of Rosario,
Rosario, Argentina
Human Virology Group
Suipacha 590
Rosario, Santa Fe 2000
Argentina

Re: Spectrum02047-23 (Identification and Characterization of Novel Alphacoronaviruses in *Tadarida brasiliensis* (Chiroptera, Molossidae) from Argentina: Insights into Recombination as a Mechanism Favoring Bat Coronavirus Cross-Species Transmission)

Dear Prof. Adriana A Giri:

Thank you for submitting your manuscript to Microbiology Spectrum. I would like to consider a minor revision of this manuscript. When submitting the revised version of your paper, please provide (1) point-by-point responses to the issues raised by the reviewers as file type "Response to Reviewers," not in your cover letter, and (2) a PDF file that indicates the changes from the original submission (by highlighting or underlining the changes) as file type "Marked Up Manuscript - For Review Only". Please use this link to submit your revised manuscript - we strongly recommend that you submit your paper within the next 60 days or reach out to me. Detailed instructions on submitting your revised paper are below.

Link Not Available

Sincerely,

Biao He

Journals Department
Reviewer comments:

Reviewer #2 (Comments for the Author):

Cerri et al present a study identifying novel alphacoronaviruses in *Tadarida brasiliensis* bats in Argentina. This is a well conducted study with clearly presented results but I have some questions regarding the recombination analysis:

1) Did the authors consider using RDP5 for the analysis? This updated version of the software has been available since 2021 and has considerably improved the verification and interpretation of recombination events that are detected. This is an important step as RDP does frequently call recombination events that do not stand up to additional analysis. While the authors have

conducted phylogenetic analysis that somewhat backs up this as a true event, it would be interesting to see if the updated software also identifies the event.

2) Did the authors conduct any manual manipulation of the positions of the recombination breakpoints? Again, RDP does not always call the breakpoint accurately. Looking at Figure 4 the contribution of sequence from the minor parent (the novel virus) looks like it may extend into orf1b. Do the authors consider this to be a possibility and if so how does the increased sequence alter the phylogenetic analysis conducted? Recombination junctions have been demonstrated to occur more frequently at gene boundaries, and this increased region would fit with such previous findings.

Minor comment:

1) In Figure 5 the major and minor colour labels appear to have been written the wrong way round in the figure legend.

Reviewer #5 (Comments for the Author):

This work contributes to the body of knowledge surrounding coronaviruses. This paper makes break-throughs in discovering novel coronaviruses in bats from Argentina.

This manuscript was well written and was very detail oriented. The phylogenetic trees are easy to interpret and the analysis was very robust.

Overall, this is a fantastic paper that contributes to the field of virology.

Staff Comments:

Preparing Revision Guidelines

Please return the manuscript within 60 days; if you cannot complete the modification within this time period, please contact me. If you do not wish to modify the manuscript and prefer to submit it to another journal, please notify me of your decision immediately so that the manuscript may be formally withdrawn from consideration by Microbiology Spectrum.

Response to reviewers comments (Manuscript ID: Spectrum02047-23R1)

Reviewer #2 (Comments for the Author):

Cerri et al present a study identifying novel alphacoronaviruses in *Tadarida brasiliensis* bats in Argentina. This is a well conducted study with clearly presented results but I have some questions regarding the recombination analysis:

1) Did the authors consider using RDP5 for the analysis? This updated version of the software has been available since 2021 and has considerably improved the verification and interpretation of recombination events that are detected. This is an important step as RDP does frequently call recombination events that do not stand up to additional analysis. While the authors have conducted phylogenetic analysis that somewhat backups this as a true event, it would be interesting to see if the updated software also identifies the event.

Authors: We appreciated the reviewer suggestion.

The RDP4 version was used because it is the latest stable version of the programme. We are aware of the improvements in RDP5, particularly in handling large datasets. However, we did not use it because the software is still in Beta version. Nevertheless, as suggested, we ran the same dataset alignment using RDP5, obtaining similar results to those revealed by RDP4. For the particular case of the recombination event shown in Figure 4, RDP5 identifies the same event, being the recombinant region smaller and the breakpoints falling inside the previous 1,618 bp reported region, which was supported also by phylogenetic incongruence analysis. In conclusion both software versions recognize the event with the same number of methods and similar p-values.

2) Did the authors conduct any manual manipulation of the positions of the recombination breakpoints? Again, RDP does not always call the breakpoint accurately. Looking at Figure 4 the contribution of sequence from the minor parent (the novel virus) looks like it may extend into orf1b. Do the authors consider this to be a possibility and if so how does the increased sequence alter the phylogenetic analysis conducted? Recombination junctions have been demonstrated to occur more frequently at gene boundaries, and this increased region would fit with such previous findings.

Authors: We did not manually manipulate the recombination breakpoints in the original submission.

We agree with Reviewer #2 that it may be beneficial for the virus fitness to transfer regions at gene boundaries. However, it is frequent for some coronaviruses transferring small subsections of S1 subdomain (De Klerk 2022; DOI: <https://doi.org/10.1093/ve/veac054>) as well. Nevertheless, due to the reviewer's observation we realized the genomic map in Figure 4 (original submission) was out of scale and was corrected accordingly in the revised version of the manuscript. Besides, the graphic was improved by relocating the genomic map in the top of RDP plot, adding the S1 and S2 sites and including the limits of both the recombinant region and the S gene with vertical lines for clarity. Figure 4 legend was updated accordingly as follows:

Original manuscript (lines 438 to 439, page 15): "The region between the recombination breakpoints is highlighted in the Tb1 genomic map".

Marked-up manuscript (lines 742 to 744, page 22): "The region between the recombination breakpoints is shaded and limited by dashed lines in the Tb1 genomic map. Limits of the Spike gene and S1/S2 subdomains are also shown."

In addition, and looking at the new Figure 4 (bottom, left), your observation is relevant since the putative recombinant region seems to be more extensive than the 1,618 bp fragment identified by RDP (positions 21,796 - 23,414 bp relative to the recombinant sequence). To verify your observation, we performed phylogenetic trees increasing the length of the putative recombinant region (region 2) from positions 20,687- 23,414 bp in order to include the beginning of the S gene. An illustrative tree including the extended region is depicted in Figure 4 bis (bottom, right) and shows that the expected changes in the topology of the phylogenetic tree incongruence analysis indicating recombination were not observed. Therefore, we conclude that the 1,618 bp fragment originally detected by the RDP program better represents the putative recombination event.

Figure 4 (Recombinant 1,618 bp fragment)

Figure 4bis (extended Recombinant region)

Minor comment:

1) In Figure 5 the major and minor color labels appear to have been written the wrong way round in the figure legend.

Authors: Reviewer #2 observation is correct. Additionally, we detect a mistake in the positions of Regions 1 to 3 that were not referred to the recombinant strain (OL410609).

Accordingly, Figure 5 legend was corrected as follows.

Original manuscript (lines 442 to 445, page 15): “Maximum likelihood trees from REGION 1 (nt positions 1–21,941), REGION 2 (nt positions 21,942–23,560), and REGION 3 (nt positions 23,561–28,348). The recombinant strain is indicated in bold; the major and minor parents are indicated in purple and blue, respectively.”

Marked-up manuscript (lines 747 to 750, page 23) as: “Maximum likelihood trees from REGION 1 (nt positions 1–21,795), REGION 2 (nt positions 21,796 – 23,414), and REGION 3 (nt positions 23,415–28,340). The recombinant strain is indicated in bold; the major and minor parents are indicated in blue and purple, respectively”.

Accordingly, the positions in the text body were corrected as follows.

Specifically it says (original submission; Line 425, page 15): “The recombinant segment extended between 21,942 and 23,560 bp...”

It was corrected (marked-up manuscript; line 384, page 13) as: “The recombinant segment extended between 21,796 and 23,414 bp...”

Reviewer #5 (Comments for the Author):

This work contributes to the body of knowledge surrounding coronaviruses. This paper makes breakthroughs in discovering novel coronaviruses in bats from Argentina.

This manuscript was well written and was very detail oriented. The phylogenetic trees are easy to interpret and the analysis was very robust.

Overall, this is a fantastic paper that contributes to the field of virology.

Authors: we are grateful to the comments and considerations of Reviewer #5.

#Corrections made by the authors

Original submission (Line 372, page 13): “The two trees were rooted using BetaCoVs as outgroups.” was corrected to “All trees were rooted using BetaCoVs as outgroups.” in the marked-up manuscript (Line 722, page 22).

July 14, 2023

Prof. Adriana A Giri
Human Virology Group, National Council of Scientific and Technical Research, Institute of Molecular and Cell Biology of Rosario,
Rosario, Argentina
Human Virology Group
Suipacha 590
Rosario, Santa Fe 2000
Argentina

Re: Spectrum02047-23R1 (Identification and Characterization of Novel Alphacoronaviruses in Tadarida brasiliensis (Chiroptera, Molossidae) from Argentina: Insights into Recombination as a Mechanism Favoring Bat Coronavirus Cross-Species Transmission)

Dear Prof. Adriana A Giri:

Thank you for revising the manuscript. I am glad to inform you that your manuscript has been accepted, and I am forwarding it to the ASM Journals Department for publication. You will be notified when your proofs are ready to be viewed.

Sincerely,

Biao He
Editor, Microbiology Spectrum
